# Optimizing One-Sample Tests for Proportions in Single- and Two-Stage Oncology Trials

**DOI:** 10.3390/cancers17152570

**Published:** 2025-08-04

**Authors:** Alan David Hutson

**Affiliations:** Roswell Park Comprehensive Cancer Center, Department of Biostatistics and Bioinformatics, Elm and Carlton Streets, Buffalo, NY 14623, USA; alan.hutson@roswellpark.org

**Keywords:** perturbation test, exact binomial test, small-sample power, clinical trial

## Abstract

Phase II oncology trials often use single-arm designs when randomized trials are too expensive or impractical, such as in rare diseases. These trials typically test whether a treatment’s success rate exceeds a specified benchmark. Standard statistical methods, like the exact binomial test or Simon’s two-stage design, are commonly used but tend to be conservative, often underestimating the actual probability of incorrectly rejecting a true null hypothesis (Type I error). To address this, a new method is proposed that blends the binomial distribution with simulated normal data to create an unbiased estimate of treatment success. This convolution-based method improves the precision of Type I error control and can lead to more efficient trial designs. It also introduces a new two-stage design that includes an early stopping point for futility, offering flexibility and reduced sample sizes without compromising statistical rigor. Compared to traditional methods, this approach can lower the cost and shorten the duration of trials, making it a promising tool for early-stage oncology research.

## 1. Introduction

Common designs for phase II oncology trials typically focus on testing the hypothesis $H_{0}:\pi =\pi_{0}$ versus $H_{a}:\pi >\pi_{0}$ in a one-arm non-randomized setting. Although randomized trials are generally preferred, considerations such as cost, feasibility, and the rarity of certain cancer types often necessitate the use of single-arm designs. The estimated per-patient cost of conducting an oncology clinical trial was approximately $59,500, as reported by Batelle in 2013 [1]. More recent studies, particularly those involving cellular therapies, report substantially higher costs, in some cases exceeding $500,000 per treatment cycle [2,3,4]. Consequently, there is a critical need to optimize both phase II and phase III randomized trial designs to shorten trial duration and reduce required sample sizes. These efforts not only address escalating costs but also aim to expedite the availability of effective therapies to cancer patients. The focus of this work is towards optimizing phase II one-arm oncology trials with a binary endpoint in terms of reduced sample size and increased efficiency.

Commonly employed binary endpoints in phase II trials include objective response, complete response, and progression-free, event-free, or overall survival at fixed time points, such as 6 months or 1 year. A key feature of non-randomized, single-arm phase II trials is that, in many cancer indications, the standard-of-care population response rate is sufficiently well-characterized to serve as a comparator. If no promising signal is observed, further development, including progression to a randomized phase II or III trial, is typically not pursued.

In single-stage designs, the hypothesis about a rate or proportion, $H_{0}:\pi =\pi_{0}$ versus $H_{a}:\pi >\pi_{0}$, is most often tested using an exact binomial test, where the Type I error rate ≤α. In contrast, one-arm two-stage designs commonly employ Simon’s two-stage design [5], using either the minimax or optimal design configurations. The minimax design minimizes the total sample size, while the optimal design minimizes the expected sample size under the null hypothesis, where with the constraints are that the Type I error rate ≤α and the Type II error rate ≤β. Both designs incorporate an interim futility analysis at a predetermined sample size to allow early termination for lack of efficacy. Historically, it is noteworthy that very similar sampling schemes were developed decades earlier in the field of quality control, referred to as double sampling plans [6], with Simon’s two-stage design representing a special case of these earlier methods [7].

The issue with both single-stage and two-stage designs is that the exact Type I error rate is often considerably lower than the desired Type I error rate α and the desired power is often larger than the desired power value 1−β, due to the discreteness of the underlying binomial distribution under both the null and alternative hypotheses for a given design. This phenomenon is illustrated in the so-called saw-tooth plots in Figure 1, which display the exact Type I error across a range of potential null values for $\pi_{0}$ with sample sizes n=10,20,30,40. As a result, these tests can be conservative in certain scenarios where the exact Type I error rate falls substantially below the target level α.

One approach to mitigate this conservatism is to incorporate a continuity correction [8]; however, this does not eliminate the saw-tooth behavior in Type I error control. Similarly, the power function may also exhibit a saw-tooth pattern and can be non-monotonic [9]. For a fixed sample size *n*, there are *n* discrete values of $\pi_{0}$ corresponding to k=1,2,…,n (k=0 is infeasible), at which the Type I error equals α, given by(1)$$\begin{array}{c} \pi_{0}=I_{\alpha }^{-1}\left(k,n-k+1\right), \end{array}$$
where $I^{-1}$ is the inverse regularized incomplete beta function.

Interestingly, Simon’s two-stage design can exhibit the same phenomenon, where the exact Type I error rate is often much lower than the derired Type I error rate α. In this case, however, we cannot produce a saw-tooth plot as a function of $\pi_{0}$ alone, since Simon’s two-stage design also depends on the choice of the alternative response rate, denoted $\pi_{1}\left(>\pi_{0}\right)$. In Figure 2, we present the exact Type I error rate and power for testing $H_{0}:\pi =\pi_{0}$ versus $H_{a}:\pi >\pi_{0}$, with $\pi_{0}=0.1$ and $\pi_{1}$ varying from 0.19 to 0.8, while fixing α=0.05 and power = 0.80 for the minimax design. As $\pi_{1}$ increases, the required total sample size decreases, thus reducing the dimensionality of the sample space for design choices. In fact, for this example, there are very few instances where the exact Type I error rate approaches the nominal α. For example, when $\pi_{1}=0.35$, the exact Type I error rate is 0.027 with a final stage sample size of n=18. Increasing $\pi_{1}$ to 0.36 raises the exact Type I error rate to 0.044, with a corresponding final stage sample size of n=14. For smaller values of *n*, the exact Type I error rate can drop as low as 0.015. Similarly, as the sample size decreases with increasing $\pi_{1}$, the exact power may deviate considerably from the target power.

In this note, we illustrate how a straightforward convolution of a binomial random variable with a simulated normal random variable enables the construction of an unbiased estimator for the rate parameter π, and facilitates inference for $H_{0}:\pi =\pi_{0}$ versus $H_{a}:\pi >\pi_{0}$ with precise Type I error control. This in turn can reduce sample size requirements for both one-stage and two-stage designs.

In Section 2, we define the convolution estimator, derive its density and distribution functions, outline key theoretical properties including its expectation and variance, and present a toy example to demonstrate the *p*-value calculation.

We then provide a detailed comparison of the new convolution-based test with the exact binomial test in terms of Type I error control and power. In Section 3, we introduce a new two-stage design with a futility stopping rule that also achieves precise Type I error control. A direct comparison between the convolution-based two-stage design and Simon’s two-stage design is presented.

In Section 4, we provide real-world examples of both one-stage and two-stage designs, demonstrating how the convolution-based approach can reduce the cost and duration of clinical trials based on published design parameters. We conclude with final remarks.

## 2. Convolution Estimator

Our approach for constructing a test of the hypothesis $H_{0}:\pi =\pi_{0}$ versus $H_{1}:\pi >\pi_{0}$, with precise Type I error control, utilizes a convolution-based method, in which synthetic continuous noise is added to discrete binomial data. Specifically, let Y∼Binomial(n,π) denote the binomial response over *n* subjects, where π is the success probability. Let X∼N(0,h) be an independent normal random variable with mean 0 and standard deviation *h*. We define the continuous variable Z=Y+X and derive its probability density and cumulative distribution functions.

The probability density function (PDF) of *Z*, denoted $f_{Z}\left(z\right)$, is obtained by convolving the binomial probability mass function of *Y* with the normal density of *X*. Since *Y* is discrete and *X* is continuous, this results in a finite mixture of normal densities:fZ(z)=1h2π∑k=0nnkπk(1−π)n−kexp−(z−k)22h2,z∈R.

Each component of the mixture is centered at k=0,1,…,n, with mixing weights given by the binomial probabilities nkπk(1−π)n−k.

Similarly, the cumulative distribution function (CDF) of *Z*, denoted $F_{Z}\left(z\right)=P\left(Z\leq z\right)$, is derived by conditioning on the values of *Y*:(2)FZ(z)=∑k=0nnkπk(1−π)n−kΦz−kh,z∈R,
where Φ(u) is the standard normal CDF defined as$$\Phi \left(u\right)=\int_{-\infty }^{u}\frac{1}{\sqrt{2\pi }}e^{-t^{2}/2}\;dt.$$

To understand the properties of the test statistic, we compare the moments of *Y* and *Z*. The expectation and variance of *Y* are given byE[Y]=nπ,Var(Y)=nπ(1−π).

Since *Y* and *X* are independent, the corresponding moments of Z=Y+X follow directly:E[Z]=E[Y+X]=E[Y]+E[X]=nπ+0=nπ,$$\mathrm{Var}\left(Z\right)=\mathrm{Var}\left(Y+X\right)=\mathrm{Var}\left(Y\right)+\mathrm{Var}\left(X\right)=n\pi \left(1-\pi \right)+h^{2}.$$

Thus, the mean of *Z* remains identical to that of *Y*, while the variance of *Z* is increased by an additive term $h^{2}$, capturing the additional variability introduced by the normal perturbation. Throughout this note, we fix the value of h=1/100. Although this results in only a modest increase in the variance of *Z* we will demonstrate that this small adjustment is sufficient to produce tests with substantially improved power while maintaining the precise Type I error level.

To test $H_{0}:\pi =\pi_{0}$ against $H_{1}:\pi >\pi_{0}$ at significance level α, we reject $H_{0}$ if Z>c, where the critical value *c* is chosen to satisfy(3)1−∑k=0nnkπ0k(1−π0)n−kΦc−kh=α.

The solution for *c* is via numerical methods. Similarly, the *p*-value is given by(4)p=1−FZ|π=π0(z)=1−∑k=0nnkπ0k(1−π0)n−kΦz−kh,
where *z* is the value of the observed convolution-based test statistic.

Under the alternative hypothesis $H_{1}:\pi =\pi_{1}>\pi_{0}$, the corresponding CDF isFZ|π=π1(z)=∑k=0nnkπ1k(1−π1)n−kΦz−kh,
and the power of the test at $\pi =\pi_{1}$ is given by(5)Power(π1)=1−∑k=0nnkπ1k(1−π1)n−kΦc−kh.

### 2.1. Toy Example

We simulated the random variables Y∼Binomial(n=20,π=0.3) and $X∼N(0,h^{2})$ with $h=\frac{1}{100}$, and computed Z=X+Y. For each run, we calculated the convolution-based *p*-value and the exact binomial *p*-value testing $H_{0}:\pi =\pi_{0}\;\mathrm{vs}.\;H_{1}:\pi >\pi_{0}.$

We see from Table 1 that the convolution-based *p*-values are consistently close to the exact binomial *p*-values, but tend to be slightly smaller due to the smoothing effect introduced by the normal perturbation *X*. The addition of this small normal noise transforms the discrete binomial distribution into a continuous mixture distribution, which leads to slightly different tail probabilities. For runs with larger observed binomial counts *y*, both the convolution and exact tests yield smaller *p*-values, indicating stronger evidence against the null hypothesis $H_{0}:\pi =\pi_{0}$.

### 2.2. Convolution Approach and Exact Binomial Test Comparison

Table 2 presents a Type I error control and power comparison between the convolution-based test and the exact binomial test for a sample size of n=10, across varying null hypotheses $\pi_{0}\in \left\{0.1,0.2,0.3,0.4,0.5\right\}$ and corresponding alternatives $\pi_{1}\in \left\{\pi_{0},\pi_{0}+0.1,\ldots ,\min\left(\pi_{0}+0.4,1\right)\right\}$. For each $\pi_{0}$, a critical value *c* was determined so that the convolution-based test controls the Type I error exactly at level α=0.05, as defined in Equation (Equation 3). The corresponding power was computed using Equation (Equation 5), noting that power equals the Type I error when $\pi_{0}=\pi_{1}$. In contrast, the exact binomial rejection threshold *k* was chosen to ensure that the Type I error remained strictly below α.

Table 3 reports analogous results for a larger sample size of n=20. As expected, power increases for both methods with larger *n*. Across both settings, the convolution-based test consistently achieves the nominal Type I error and provides greater power than the exact binomial test, particularly when the exact binomial test is overly conservative relative to Type I error control. This improved performance is attributed to the smoothing introduced by the normal perturbation, which results in a more refined and responsive rejection region. These findings underscore the practical utility of the convolution-based test in discrete-data settings, especially when measurement or process noise is present and sample sizes are limited. The only time the classic test and the convolution based test are equivalent are the *n* values for $\pi_{0}$ where Equation (Equation 1) is satisfied.

## 3. Two Stage Design

Similar to the one-stage test, we construct a two-stage test of the hypothesis $H_{0}:\pi =\pi_{0}$ versus $H_{1}:\pi >\pi_{0}$, offering precise control of the Type I error rate and allowing for early stopping due to futility. Let the total sample size be $n=n_{1}+n_{2}$. After the first $n_{1}$ subjects have completed their endpoint assessments, a futility stopping rule is applied.

As in the one-stage design, let $Z_{1}$ denote the observed value of the convolution-based test statistic after the first $n_{1}$ subjects, and let $Z_{2}$ be the corresponding statistic based on an independent second cohort of $n_{2}$ subjects. Under the null hypothesis $H_{0}$, define the *p*-values as$$p_{1}=1-F_{Z|\pi =\pi_{0}}\left(Z_{1}\right),\;p_{2}=1-F_{Z|\pi =\pi_{0}}\left(Z_{2}\right),$$
where $p_{1}$ and $p_{2}$ follow a uniform U(0,1) distribution by the probability integral transform. The cumulative distribution function $F_{Z|\pi =\pi_{0}}$ is defined in Equation (Equation 4).

Our approach uses a *p*-value threshold at the interim analysis and applies Stouffer’s weighted z-score method for the final efficacy analysis [10]. Define the transformed statistics:$$T_{1}=\Phi^{-1}\left(p_{1}\right),\;\mathrm{based}\;\mathrm{on}\;\mathrm{the}\;\mathrm{first}\;n_{1}\;\mathrm{subjects},$$$$T_{2}=\Phi^{-1}\left(p_{2}\right),\;\mathrm{based}\;\mathrm{on}\;\mathrm{the}\;\sec\mathrm{ond}\;n_{2}\;\mathrm{subjects},$$
where $\Phi^{-1}$ is the quantile function of the standard normal distribution.

The combined test statistic is given by the following:$$T=\frac{w_{1}T_{1}+w_{2}T_{2}}{\sqrt{w_{1}^{2}+w_{2}^{2}}},$$
where the weights are defined as $w_{1}=n_{1}/n$ and $w_{2}=n_{2}/n$. With this weighting scheme, the statistic *T* follows a standard normal distribution under $H_{0}$, i.e., T∼N(0,1).

The interim futility rule is to terminate the study early if $p_{1}>p_{c}$, where the threshold $p_{c}$ may be specified by the user or selected to optimize statistical power or minimize the expected sample size,$$\mathrm{ESN}=n_{1}p_{c}+n_{2}\left(1-p_{c}\right).$$

If the study does not stop early for futility, the final *p*-value is computed as p=Φ(T), where Φ denotes the standard normal CDF. The null hypothesis $H_{0}$ is rejected if $p<\alpha^{\prime }$, where $\alpha^{\prime }$ is adjusted to ensure that the overall Type I error rate is controlled at the nominal level α.

To determine $\alpha^{\prime }$, let $a=\Phi^{-1}\left(p_{c}\right)$. The conditional probability of rejecting $H_{0}$ given that the futility boundary is not crossed is:$$P\left(T<c|T_{1}<a\right)=\frac{1}{\Phi (a)}\int_{-\infty }^{a}\Phi \left(\sqrt{2}c-x\right)\;\phi \left(x\right)\;dx,$$
where ϕ is the standard normal density function. To ensure the overall Type I error rate is α, we solve for *c* in the equation:$$p_{c}\cdot P\left(T<c|T_{1}<a\right)=\alpha ,$$
and define the adjusted significance level as $\alpha^{\prime }=\Phi \left(c\right)$, which satisfies $\alpha^{\prime }>\alpha$. Once $\alpha^{\prime }$ is determined numerically power can be calculated under $H_{1}$ via simulation. For a fixed *n* we can find combinations of $n_{1}$ and $n_{2}$ such that the overall α=0.05 and power is greater than or equal to the desired power. Practically speaking one can start at the *n* determined by the Simon two-stage design and reduce by increments of 1 until the power constraint is no longer satisfied. The search is over values of $n_{1}$ and $p_{c}$, with $n_{2}=n-n_{1}$. This will be illustrated in the next section.

### Convolution Approach and Simon Two-Stage Design Comparison

There is no straightforward way to directly compare the convolution-based approach with Simon’s two-stage design. A key distinguishing feature is that the convolution-based method precisely controls the Type I error rate, whereas Simon’s two-stage design may be conservative in some scenarios, as illustrated earlier in Figure 2. In settings where the actual Type I error of Simon’s design falls well below the nominal level, the convolution-based method can achieve the same desired power with a smaller sample size, particularly when the difference between $\pi_{1}$ and $\pi_{0}$ is substantial.

To illustrate, we consider testing $H_{0}:\pi =\pi_{0}$ versus $H_{1}:\pi >\pi_{0}$ with $\pi_{0}=0.1$ and $\pi_{1}$ ranging from 0.3 to 0.6 in increments of 0.1. The results of various Simon two-stage designs are presented in Table 4, showing the corresponding total sample size, exact Type I error, power, expected sample size (EN_0_), and probability of early stopping.

Similarly, Table 5 displays results from the convolution-based two-stage designs across the same values of $\pi_{1}$. For each design, we considered a range of $p_{c}$ values from 0.2 to 0.7 in steps of 0.01, and we report the total sample size *n*, stage-wise sample sizes $n_{1}$ and $n_{2}$, $p_{c}$, power, expected sample size (ESN), adjusted $\alpha^{\prime }$, for testing at the second stage, and the probability of early stopping, which is equal to $1-p_{c}$. The designs in Table 5 represent a subset of possible scenarios to provide a concise summary.

The convolution-based approach demonstrates a clear advantage by achieving slightly smaller sample sizes across all settings. Moreover, it is not constrained by the range of early stopping probabilities observed in the Simon designs (approximately 0.549 to 0.810 in our example). Instead, the convolution-based method allows the user to specify any desired $p_{c}$ (and thus early stopping probability), offering greater flexibility to tailor the design to the specific goals and constraints of a given clinical trial.

## 4. Real World Examples

### 4.1. One-Stage Designs

#### 4.1.1. Example 1


Our first example [11] is from a study of patients with concomitant advanced non-small cell lung cancer (NSCLC) and interstitial lung disease (ILD). This prospective, multicenter, single-arm phase 2 trial investigated the efficacy and safety of albumin-bound paclitaxel (nab-paclitaxel) in combination with carboplatin in patients with both advanced NSCLC and ILD. The primary endpoint was the overall response rate (ORR), testing $H_{0}:\pi =\pi_{0}$ versus $H_{1}:\pi >\pi_{0}$, with $\pi_{0}=0.2$, alternative $\pi_{1}=0.4$, α=0.05, and power = 0.80. Based on an exact binomial test, this required a sample size of n=35. The actual study enrolled n=36 subjects between April 2014 and September 2017, corresponding to an average accrual rate of approximately 1.3 subjects per month.

Using the same design parameters, the convolution-based test would require n=32 subjects to achieve a power of 0.8117, or n=31 for a power of 0.7967, potentially reducing the trial duration by 4 to 5 months with the corresponding cost savings. The *p*-value was <0.001 under both the exact binomial and convolution-based approaches. If a prorated response of 17 out of 31 positive responses were observed, the *p*-value using the convolution-based method would still be <0.001.

#### 4.1.2. Example 2

Our next example study [12] enrolled n=15 metastatic prostate cancer patients with AR-V7-expressing circulating tumor cells into a prospective phase II trial. The primary endpoint was PSA response, with hypothesis testing conducted as $H_{0}:\pi =\pi_{0}$ versus $H_{1}:\pi >\pi_{0}$, where $\pi_{0}=0.05$, $\pi_{1}=0.264$, α=0.10, and target power = 0.80. The true Type I error rate for the exact binomial test was 0.0362. A positive outcome in the study was thus defined as ≥3 of 15 patients achieving a PSA response.

Using the same design parameters, the convolution-based test would require n=11 subjects to achieve a power of 0.824, or n=10 for a power of 0.793. In the actual trial, 2 out of 15 subjects achieved a PSA response, yielding a *p*-value of 0.14 using the exact binomial test. The convolution-based test yielded a *p*-value of 0.106. Although not statistically significant at the α=0.10 level, this result demonstrates the relative efficiency of the convolution-based approach.

### 4.2. Two-Stage Designs

#### 4.2.1. Example 1

Our next example [13] is based on a study evaluating vinorelbine in advanced non-small cell lung cancer (NSCLC) patients aged 70 years or older. The study employed a multicenter, two-stage phase II design following Simon’s optimal method. The primary endpoint was objective response rate (ORR), with hypothesis testing structured as $H_{0}:\pi =\pi_{0}$ versus $H_{1}:\pi >\pi_{0}$, where $\pi_{0}=0.10$, $\pi_{1}=0.25$, h=0.01, α=0.05, and target power = 0.80.

The final sample size under Simon’s design was n=43 with an interim futility analysis planned at $n_{1}=18$ subjects. The actual Type I error rate achieved was 0.048. Table 6 presents three example two-stage designs generated using the convolution-based estimator. Notably, if one is willing to delay the interim futility analysis beyond $n_{1}=18$, the total required sample size can be reduced from n=43 to n=35, offering a more efficient design compared to Simon’s optimal design alternative. Even maintaining the interim analysis at $n_{1}=18$, the convolution-based approach can still reduce the total sample size to n=37.

In the observed trial data, 5 of the initial 18 subjects achieved a positive ORR, with a total of 10 ORR responses observed out of 43 by the end of the study. Simon’s optimal design specified early stopping for futility if 3 or fewer ORR responses were observed in the first stage, and declared efficacy if 8 or more total responses were observed at the second stage.

For each convolution-based design in Table 6, we retrospectively aligned the observed data to the alternative designs to estimate what would have occurred under those configurations:For n=35, $n_{1}=27$: Assuming 7 out of 27 responses, the futility *p*-value was 0.011, below the stopping threshold $\pi_{c}=0.23$. The final-stage *p*-value, assuming 8 out of 35 total responses, was 0.0158, significant at the adjusted level $\alpha^{\prime }=0.062$.For n=36, $n_{1}=24$: Assuming 6 out of 24 responses, the futility *p*-value was 0.002, below $\pi_{c}=0.24$. The final-stage *p*-value based on 8 of 36 responses was 0.0162, also significant at $\alpha^{\prime }=0.061$.For n=37, $n_{1}=18$: Assuming 5 out of 18 responses, the futility *p*-value was 0.015, which is less than $\pi_{c}=0.56$. The final *p*-value with 8 of 37 responses was 0.020, significant at $\alpha^{\prime }=0.051$.

This example again highlights the potential cost savings and efficiency gains achievable with the convolution-based approach compared to Simon’s two-stage design, while maintaining rigorous Type I error control and statistical power.

#### 4.2.2. Example 2

Our next example [14] comes from a publication describing the LuDO-N trial, a phase II, open-label, multicenter, single-arm, two-stage clinical trial in children with high-risk neuroblastoma, utilizing an alternative administration schedule of Lutetium DOTATATE. The primary endpoint is the response rate, assessed by the Revised International Neuroblastoma Response Criteria one month after completion of therapy. Hypothesis testing is structured as $H_{0}:\pi =\pi_{0}$ versus $H_{1}:\pi >\pi_{0}$, where $\pi_{0}=0.2$, $\pi_{1}=0.4$,h=0.01, α=0.1, and the target power is 0.80.

The proposed design follows Simon’s Two-Stage Minimax design. Recruitment is expected to be completed within 3–5 years. Based on the specified parameters, the Simon design requires a total sample size of n=24, with a first-stage futility analysis at $n_{1}=14$, yielding a true Type I error rate of 0.0874.

In contrast, an alternative convolution-based design would require a total of n=19 subjects, with $n_{1}=16$ in the first stage, using a futility threshold of $p_{c}=0.21$ and a final adjusted significance level of $\alpha^{\prime }=0.158$. If the projected recruitment rate is 24 subjects over 5 years (approximately 4.8 subjects per year), the convolution-based design would reduce the total accrual period by roughly one year.

## 5. Conclusions

The convolution-based approach presented in this work offers a flexible and efficient alternative to traditional exact methods for designing and analyzing single-arm phase II oncology trials with binary endpoints. By convolving the binomial distribution with a simulated normal random variable, this method produces an unbiased estimator for the response rate π and achieves precise Type I error control. This leads to reduced sample size requirements in both one-stage and two-stage designs, while maintaining desirable operating characteristics.

A significant advantage of the convolution framework is its adaptability. It can be easily modified to incorporate an interim analysis for either futility or combined efficacy and futility, allowing for early decision-making and further optimization of trial resources. Moreover, the convolution-based method provides a smooth and continuous approximation to the binomial tail distribution, making it especially valuable in scenarios where measurement error or process variability exists, and a continuous *p*-value function is preferred.

Overall, this approach enhances the efficiency of early-phase oncology clinical trial design and provides a theoretically sound and practically implementable alternative to exact binomial and Simon’s two-stage methods. It holds particular promise for high-cost therapeutic areas, such as oncology and cellular therapies, where efficient trial execution is critical.

## Figures and Tables

**Figure 1 cancers-17-02570-f001:**
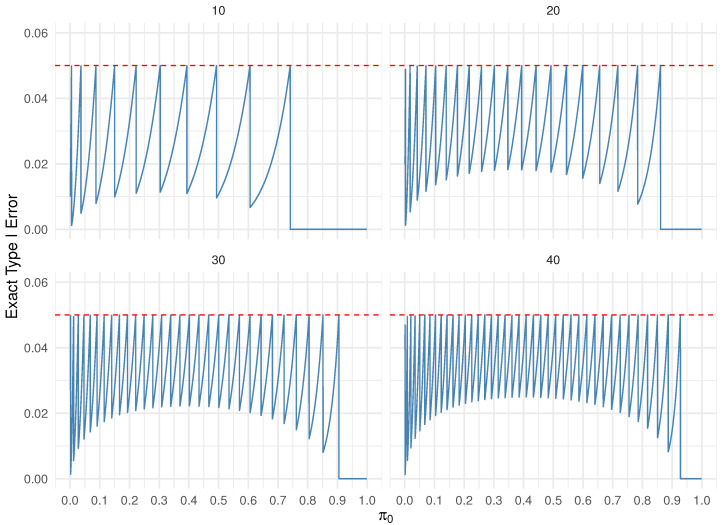
Type I error control for exact binomial test as a function of $\pi_{0}$ values, n=10,20,30,40.

**Figure 2 cancers-17-02570-f002:**
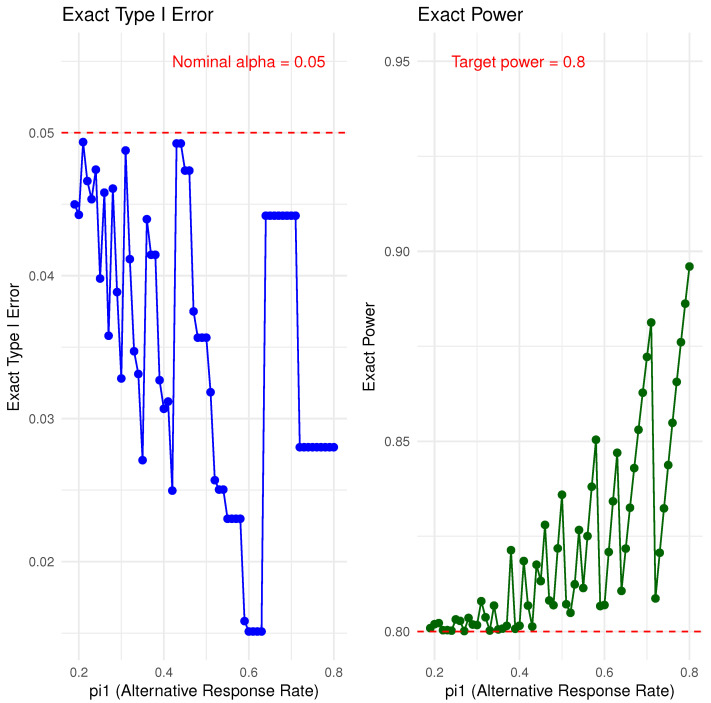
Type I error control and power for the mimimax Simon’s two-stage design as a function of $\pi_{0}=0.1$ and $\pi_{1}=0.19$ to 0.80.

**Table 1 cancers-17-02570-t001:** Comparison of mixture-based and exact binomial *p*-values over three simulations.

Run	*y*	*x*	z=x+y	$F_{Z}\left(z\right)$	Convoution *p*-Value	Exact Binomial *p*-Value
1	4	0.007968	4.007968	0.5832	0.4168	0.5886
2	11	0.014024	11.01402	0.9999	0.0001	0.0006
3	3	0.008362	3.008362	0.3701	0.6299	0.7939

**Table 2 cancers-17-02570-t002:** Comparison of power between the mixture-based convolution test and the exact binomial test for sample size n=10, across values of $\pi_{0}$ and $\pi_{1}$, h=0.01. The rejection threshold *k* ensures the binomial test controls Type I error strictly below α=0.05.

$\pi_{0}$	$\pi_{1}$	Critical *c*	Mixture Power	Rejection *k*	Binomial Power
0.1	0.1	2.9962	0.050000	4	0.012795
0.1	0.2	2.9962	0.251377	4	0.120874
0.1	0.3	2.9962	0.523352	4	0.350389
0.1	0.4	2.9962	0.757080	4	0.617719
0.1	0.5	2.9962	0.904088	4	0.828125
0.2	0.2	4.0086	0.050000	5	0.032793
0.2	0.3	4.0086	0.189362	5	0.150268
0.2	0.4	4.0086	0.415895	5	0.366897
0.2	0.5	4.0086	0.663109	5	0.623047
0.2	0.6	4.0086	0.855538	5	0.833761
0.3	0.3	5.0195	0.050000	6	0.047349
0.3	0.4	5.0195	0.171407	6	0.166239
0.3	0.5	5.0195	0.383292	6	0.376953
0.3	0.6	5.0195	0.638272	6	0.633103
0.3	0.7	5.0195	0.852383	6	0.849732
0.4	0.4	6.9878	0.050000	8	0.012295
0.4	0.5	6.9878	0.158735	8	0.054688
0.4	0.6	6.9878	0.358174	8	0.167290
0.4	0.7	6.9878	0.619691	8	0.382783
0.4	0.8	6.9878	0.856551	8	0.677800
0.5	0.5	7.9876	0.050000	9	0.010742
0.5	0.6	7.9876	0.154390	9	0.046357
0.5	0.7	7.9876	0.357879	9	0.149308
0.5	0.8	7.9876	0.645587	9	0.375810
0.5	0.9	7.9876	0.909147	9	0.736099

**Table 3 cancers-17-02570-t003:** Power comparison between the mixture-based convolution test and the exact binomial test for sample size n=20, across values of $\pi_{0}$ and $\pi_{1}$, h=0.01. The rejection threshold *k* ensures the binomial test controls Type I error strictly below α=0.05.

$\pi_{0}$	$\pi_{1}$	Critical *c*	Mixture Power	Rejection *k*	Binomial Power
0.1	0.1	4.0143	0.050000	5	0.043174
0.1	0.2	4.0143	0.386941	5	0.370352
0.1	0.3	4.0143	0.772408	5	0.762492
0.1	0.4	4.0143	0.951708	5	0.949048
0.1	0.5	4.0143	0.994442	5	0.994091
0.2	0.2	7.0045	0.050000	8	0.032143
0.2	0.3	7.0045	0.281501	8	0.227728
0.2	0.4	7.0045	0.638410	8	0.584107
0.2	0.5	7.0045	0.892613	8	0.868412
0.2	0.6	7.0045	0.983738	8	0.978971
0.3	0.3	9.0186	0.050000	10	0.047962
0.3	0.4	9.0186	0.249643	10	0.244663
0.3	0.5	9.0186	0.593093	10	0.588099
0.3	0.6	9.0186	0.874692	10	0.872479
0.3	0.7	9.0186	0.983230	10	0.982855
0.4	0.4	11.9910	0.050000	13	0.021029
0.4	0.5	11.9910	0.229635	13	0.131588
0.4	0.6	11.9910	0.562559	13	0.415893
0.4	0.7	11.9910	0.865636	13	0.772272
0.4	0.8	11.9910	0.985944	13	0.967857
0.5	0.5	13.9918	0.050000	15	0.020695
0.5	0.6	13.9918	0.224232	15	0.125599
0.5	0.7	13.9918	0.568302	15	0.416371
0.5	0.8	13.9918	0.890702	15	0.804208
0.5	0.9	13.9918	0.995777	15	0.988747

**Table 4 cancers-17-02570-t004:** Summary of Simon Two-Stage Designs with one-sided Type I error rate α=0.05, Power = 0.8, $\pi_{0}=0.1$, h=0.01. Designs are shown for various response probabilities $\pi_{1}$.

$\pi_{1}$	*n*	$n_{1}$	$r_{1}$	$r_{2}$	Type I Error	Power	EN_0_	P (Early Stop)	Method
0.3	25	15	1	5	0.033	0.802	19.5	0.549	Minimax
0.3	26	12	1	5	0.036	0.805	16.8	0.659	
0.3	27	11	1	5	0.040	0.806	15.8	0.697	
0.3	29	10	1	5	0.047	0.805	15.0	0.736	Optimal
0.4	13	8	1	3	0.031	0.802	8.9	0.813	Minimax
0.4	15	4	0	3	0.043	0.818	7.8	0.656	Optimal
0.5	8	4	0	2	0.036	0.836	5.4	0.656	Minimax
0.5	9	3	0	2	0.041	0.828	4.6	0.729	Optimal
0.6	6	3	0	2	0.015	0.807	3.8	0.729	Minimax
0.6	8	2	0	2	0.025	0.819	3.1	0.810	Optimal

**Table 5 cancers-17-02570-t005:** Summary of Convolution-Based Two-Stage Designs with one-sided Type I error rate α=0.05, Power = 0.8, $\pi_{0}=0.1$, h=0.01. Designs are shown for various response probabilities $\pi_{1}$.

$\pi_{1}$	*n*	$n_{1}$	$n_{2}$	$p_{c}$	Power	ESN	$\alpha^{\prime }$	P (Early Stop)
0.3	23	16	7	0.34	0.810	18.4	0.055	0.66
0.3	23	17	6	0.32	0.807	18.9	0.056	0.68
0.3	23	17	6	0.30	0.806	18.8	0.057	0.70
0.3	23	15	8	0.31	0.806	17.5	0.056	0.69
0.3	23	14	9	0.37	0.806	17.3	0.054	0.63
0.3	23	14	9	0.45	0.806	18.0	0.052	0.55
0.3	23	14	9	0.46	0.806	18.1	0.052	0.54
0.3	23	16	7	0.69	0.806	20.8	0.050	0.31
0.3	23	16	7	0.36	0.805	18.5	0.054	0.64
0.3	23	15	8	0.37	0.805	18.0	0.054	0.63
0.3	23	13	10	0.50	0.805	18.0	0.051	0.50
0.3	23	13	10	0.53	0.805	18.3	0.051	0.47
0.3	23	12	11	0.70	0.805	19.7	0.050	0.30
0.4	11	8	3	0.20	0.801	8.6	0.066	0.80
0.4	11	9	2	0.21	0.801	9.4	0.064	0.79
0.5	7	5	2	0.20	0.809	5.4	0.066	0.80
0.5	7	5	2	0.25	0.805	5.5	0.060	0.75
0.5	7	5	2	0.30	0.801	5.6	0.057	0.70
0.6	5	3	2	0.29	0.810	3.6	0.057	0.71
0.6	5	3	2	0.38	0.804	3.8	0.053	0.62
0.6	5	3	2	0.40	0.801	3.8	0.053	0.60
0.6	5	3	2	0.52	0.801	4.0	0.051	0.48

**Table 6 cancers-17-02570-t006:** Convolution-based design scenarios for Example 1: Two-stage design.

*n*	$n_{1}$	$n_{2}$	$\pi_{c}$	Power	ESN	$\alpha^{\prime }$
35	27	8	0.23	0.803	28.8	0.062
36	24	12	0.24	0.802	26.9	0.061
37	18	19	0.56	0.801	28.6	0.051

## Data Availability

The data utilized to illustrate this work is provided in the body of the text in Section 4.

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
