# Peer review of "Optimizing One-Sample Tests for Proportions in Single- and Two-Stage Oncology Trials"

_cancers, 2025, doi:10.3390/cancers17152570_

Round 1

Reviewer 1 Report

Comments and Suggestions for Authors

I really enjoyed reading this paper.  I think the idea is very, very clever and a nice way of using convolution/perturbation.  

I couldn't find the Figures (may be my issue in not knowing where to find them in the system)

I think the examples provide a nice articulation of the efficiency gains with this method.  I would expect there do exist some potential examples where the gain is more modest - I could be helpful to provide an example for readers' intuition.  

Are there R or other code available that could be shared for folks to implement this straightforwardly?

Perhaps supplemental, but I thought it could be useful to provide a figure that visually summarizes how this method "smooths" out the saw-tooth discrete properties.  

It would be worth commenting (but no need to provide full summaries) of how robust this to to selection of h.  I think 1/100 is very reasonable, but also since it is arbitrary it might be worth commenting on how different values really minimally impact properties.  

Author Response

Reviewer 1

Critique 1. I couldn't find the Figures (may be my issue in not knowing where to find them in the system)

Response: The plots were on Page 18 and 19. They were not in a separate file.

Critique 2. I think the examples provide a nice articulation of the efficiency gains with this method.  I would expect there do exist some potential examples where the gain is more modest - I could be helpful to provide an example for readers' intuition. 

Response: The improvement is more modest in scenarios where the condition described in the first section, specifically involving the inverse regularized incomplete beta function, is satisfied or approximately satisfied. We have added a clarifying sentence at the end of the univariate test section to reflect this point.

Critique3. Are there R or other code available that could be shared for folks to implement this straightforwardly?

Response:  For the one stage test the program is straightforward. I will add as supplemental materials.

# Parameters

n <- 36

pi0 <- 0.20

h <- 100

sig <- 1 / h

y <- 20

x <- rnorm(1, mean = 0, sd = sig)  # generate x with mean 0 and SD = sig

z <- x+y  #obs + noise

# Compute the p-value

sum <- 0

for (j in 0:n) {

  binom_prob <- dbinom(j, size = n, prob = pi0)

  normal_prob <- pnorm((z - j) / sig)

  sum <- sum + binom_prob * normal_prob

}

pvalue <- 1 - sum

# Print the results

cat("Generated x =", x, "\n")

cat("P-value for z =", z, "is", pvalue, "\n")

Critique 4. Perhaps supplemental, but I thought it could be useful to provide a figure that visually summarizes how this method "smooths" out the saw-tooth discrete properties. 

Response: The issue here is that the new method will simply be a straight line across at the level alpha across all values of π0.

Critque 5. It would be worth commenting (but no need to provide full summaries) of how robust this to to selection of h.  I think 1/100 is very reasonable, but also since it is arbitrary it might be worth commenting on how different values really minimally impact properties. 

Response: Heuristically 1/100 1/1000 1/10000 all generate the same results since there is somewhat of an automatic scaling going on.

Reviewer 2 Report

Comments and Suggestions for Authors

This manuscript considers the convolution of a Gaussian random variable with a binary random variable to get over the discrete error issue when designing binary outcome based phase II trials. 

In general, this article is very well written. The method described is simple yet impactful, techniques, empirical and real data sections are easy to follow. 

The comments below are with regards to relatively minor clarifying points. 

a) The toy example mentions h=0.01. How was this arrived at? Prior experience? Any guidance to the practioner as to how to choose h or stick to 0.01 as a default choice?

b) The p-value and power expressions seem to be monotonic with respect to h. Any recommendations on a grid to h to try and optimize?

c) Are all empirical results in table 2 and table 3 with respect to the same h of 0.01? This should be clarified in text and perhaps also in the table captions themself.

d) Some additional plots on how the disjoint sawtooth like error behavior is improved with the convolution would be helpful for the reader.

Author Response

Reviewer 2

Critique 1. The toy example mentions h=0.01. How was this arrived at? Prior experience? Any guidance to the practioner as to how to choose h or stick to 0.01 as a default choice?

Response: Heuristically 1/100 1/1000 1/10000 all generate the same results since there is somewhat of an automatic scaling going on.

Critique 2. The p-value and power expressions seem to be monotonic with respect to h. Any recommendations on a grid to h to try and optimize?

Response: Sorry, I am not sure where you are seeing this. The value h=0.01 was used throughout all calculations.

Critique 3.  Are all empirical results in table 2 and table 3 with respect to the same h of 0.01? This should be clarified in text and perhaps also in the table captions themself.

Response: Yes, h=0.01 is used throughout all calculations. I added this to all figures and examples for greater clarity.

Critique 4. Some additional plots on how the disjoint sawtooth like error behavior is improved with the convolution would be helpful for the reader.

Response: The issue here is that the new method will simply be a straight line across at the level alpha.